# The conditioned medium from mesenchymal stromal cells pretreated with proinflammatory cytokines promote fibroblasts migration and activation

Chenyang Liu[1], Chengchun Wang[2], Fengbo Yang[3], Yichi Lu[2], Pan Du[2], Kai Hu[1], Xinyao Yin[2], Peng Zhao[4]*, Guozhong Lu[4]*

1 Nanjng University of Traditional Chinese Medcine, Nanjng, Jiangsu, China, 2 Wuxi School of Medicine, Jiangnan University, Wuxi, Jiangsu, China, 3 Nantong University, Nantong, Jiangsu, China, 4 Engineering Research Center of the Ministry of Education for Wound Repair Technology, Jiangnan University, The Affiliated Hospital of Jiangnan University, Jiangsu, China

* luguozhong@hotmail.com (GL); zhaopengcn@outlook.com (PZ)

**Data Availability Statement:** All relevant data are within the paper and its Supporting information files.

## Abstract

Human dermal fibroblasts (HDFs) play important roles in all stages of wound healing. However, in nonhealing wounds, fibroblasts are prone to aging, resulting in insufficient migration, proliferation and secretion functions. Recent studies have suggested that mesenchymal stromal cells (MSCs) are conducive to wound healing and cell growth through paracrine cytokine signaling. In our studies, we found that conditioned medium of MSCs pretreated with IFN-γ and TNF-α (IT MSC-CM) has abundant growth factors associated with wound repair. Our in vitro results showed that the effects of IT MSC-CM on promoting cell migration, proliferation and activation in HDFs were better than those of conditioned medium from mesenchymal stromal cells (MSC-CM). Moreover, we embedded a scaffold material containing IT MSC-CM and reconfirmed that cell migration and activation were superior to that in the presence of MSC-CM in vivo. Generally, PDGF-BB is perceived as a promoter of the migration and proliferation of HDFs. Moreover, a high level of PDGF-BB in IT MSC-CM was detected, according to which we guess that the effect on HDFs may be mediated by the upregulation of PDGF-BB. These studies all showed the potential of IT MSC-CM to promote rapid and effective wound healing.

## Introduction

Fibroblasts, as the main cellular component of the dermis, participate in the entire process of wound repair by regulating extracellular matrix precipitation and paracrine [1]. Granulation tissue filling is the key step in the course of normal wound repair. Cytokines include fibroblast growth factor (FGF), cell adhesion molecule-1 (ICAM-1), vascular cell adhesion molecule-1 (VCAM-1), connective tissue growth factor (CTGF) and type I or III collagen secreted by fibroblasts are very important during granulation tissue formation [2]. During the course of wound repair, fibroblasts, vascular endothelial cells and stem cells in the surrounding tissues

**Funding:** This research was funded by the National Key R&D Program of China [2016YFE0204400] and the Social Development Program of Jiangsu Province [BE2018626] in the form of funds to GL, and the Scientific Research Program of Health Committee of Wuxi City in the form of funds to PZ [ZM005]. In addition, the funders had no role in study design, data collection and analysis, decision to publish, or preparation of the manuscript.

move to the wound surface in response to the chemotactic signaling, which promotes vascularization and extracellular matrix deposition to rapidly form granulation tissue in the wound [3,4]. Therefore, fibroblasts can migrate to the wound in a timely manner and can proliferate and secrete various growth factors and collagen fibers in the early stage of repair, eventually forming granulation tissue together with new capillaries to fill tissue defects and create conditions for coverage by epidermal cells, which is the key to rapid wound healing. Moreover, During the trauma remodeling phase, type I and type III collagen secreted by fibroblasts are critical in promoting the precipitation of extracellular matrix on the wound surface. And a high percentage of type III collagen results in a delicate and smooth skin tissue. In contrast, type I collagen appears in large quantities, resulting in a scar structure, with a hard and inelastic texture.

However, in the process of chronic wound healing, fibroblasts are prone to aging, resulting in insufficient migration and secretion, which leads to a persistent inflammatory response and hypovascularization, ultimately leading to nonhealing wounds [5,6]. Therefore, promoting the migration and activity of fibroblasts is one of the key points to accelerate chronic wound healing.

MSCs, as the source of cells for tissue repair, have the potential for self-renewal and differentiation and can realize functional tissue repair through self-differentiation and paracrine signaling. Studies have shown that MSCs can significantly accelerate the healing of acute wounds, diabetes ulcers, radiation ulcers and other chronic wounds in animals and humans [7,8]. Our study found that the paracrine effect of MSCs is plastic; that is, under the stimulation of a certain intensity of inflammatory factors, MSCs themselves will produce a variety of cytokines, including those capable of regulating the immune microenvironment in wounds. In addition, MSCs also produce a large amount of growth factors, such as tumor necrosis factor-induced Dutch gene-6 (TSG6), platelet-derived growth factor (PDGF), VEGF, and FGF, which can also stimulate the repair effects of endothelial cells, fibroblasts and tissue precursor cells [9–11]. Among these factors, PDGF-BB is one of the key growth factors that promote the proliferation, migration and secretion of fibroblasts [12].

However, IT MSC-CM has not been reported to have the capacity to stimulate cell migration, proliferation and activation of fibroblasts, which are essential for accelerating the wound healing process, or whether its effects are superior to those of MSC-CM. Thus, we investigated whether IT MSC-CM effectively increases the migration, proliferation and activation of fibroblasts in vivo and in vitro.

## Materials and methods

### Isolation and culture of UC-MSCs

The UC-MSCs used in our experiments were isolated from human umbilical cords as previously described [9,13]. And the Medical Ethics Committee of Affiliated Hospital of Jiangnan University approved all experimental procedures. Ethics Approval No:LS2021046. Briefly, umbilical cords from healthy, full-term deliveries (October 10, 2021) were obtained after parental consent and were transferred to a biosafety cabinet in sterile PBS within 4 hours. Gelatinous tissues without blood vessels were separated from the umbilical cord in PBS and cut into small pieces, which were then transferred to 10-cm-diameter dishes and covered with Dulbecco's modified Eagle's medium (DMEM, Gibco) supplemented with 15% FBS (Gibco), 10 ng/ml bFGF (PeproTech), and 100 mg/mL penicillin and streptomycin (Thermo) at 37°C under 5% $CO_2$. Medium replenishment was performed every 3 days, and nonadherent cells and tissues were removed after 7 days.

## Isolation and culture of HDFs

Primary HDFs were isolated from healthy male foreskin tissue with parental consent. Briefly, foreskin tissue was transferred to a biosafety cabinet in sterile PBS within 4 hours. Foreskin tissue was digested overnight with 0.1% collagenase I, and then the epidermis was separated from the dermis. The dermal tissue was washed in PBS 3 times and cut into small pieces. Then, the cells were transferred to 10-cm-diameter dishes, covered with DMEM (Gibco) supplemented with 10% FBS (Gibco) and 100 mg/mL penicillin and streptomycin (Thermo) at 37°C in a humidified atmosphere containing 5% $CO_2$ and subcultured every 3 days. Nonadherent cells and tissues were removed after 7 days.

## Identification of MSCs by flow cytometry (FCM) and identification of HDFs or myofibroblasts by immunofluorescence (IF)

A suspension of P3 generation UC-MSCs was collected, and the density was adjusted to $1\times10^6$/ml, after which the following monoclonal fluorescent antibodies were added: CD31-ECD, CD34-ECD, CD45-ECD, HLA-DR-ECD, CD29-ECD, CD90-ECD, CD73-ECD and CD105-ECD (Bio Legend). Then, the cells were incubated in the dark at 4°C for 30 min, washed twice with PBS and resuspended. The expression of related antigens was detected by flow cytometry.

HDFs were identified by IF. Furthermore, IF was also used to investigate the activation of fibroblasts after 24 hours of stimulation by each group of stem cell supernatant. Briefly, the cells were fixed with 4% paraformaldehyde for 10 minutes, permeabilized with 0.1% Triton™ X-100 for 15 minutes, and blocked with 2% BSA for 45 minutes at room temperature. The HDFs were labeled with vimentin antibody (Abcam, Alexa Fluor 594) at a 1:500 dilution in 0.1% BSA, and myofibroblasts were labeled with α-SMA antibodyat (Proteintech, Alexa Fluor 488) a 1:200 dilution in 0.1% BSA, and incubated at 4°C overnight. Nuclei were stained with DAPI.

## Preparation of conditioned medium

MSCs were cultured in 10-cm-diameter culture dishes until they reached 90% confluence and were then stimulated with IT (20 ng/mL IFN-γ and TNF-α). Twenty-four hours later, MSCs were washed with PBS to remove cytokines and were then cultured in FBS-free medium for 12 hours. Conditioned medium was collected and centrifuged at 300 x g for 5 minutes to remove cell debris, and then it was condensed by an ultrafiltration membrane at 3 kDa. The medium was then stored at -80°C.

## Evaluation of paracrine behaviors

The concentration of PDGF-BB in MSC-CM and IT MSC-CM or the concentration of bFGF in HDFs after each group was stimulated was measured by enzyme-linked immunosorbent assay according to the manufacturer's directions (MultiSciences), and 100 μL of each sample and standard was used. The absorbance (450 nm) for each sample was analyzed by a microplate reader (Cytation5, Bio Tek) and was interpolated with a standard curve.

## Scratch wound assay

The scratch wound assay was conducted as previously described [14]. Briefly, when HDFs formed a 100% confluent monolayer, a scratch was made in each culture using a disposable pipette tip (200 μl), and the cells were washed 3 times with PBS. Then, the cells were treated with DMEM, MSC-CM or IT MSC-CM. The wound area was photographed using a Motic AE

2000 inverted microscope (Motic Corporation, China) immediately and 6 h, 12 h, 24 h and 48 h after MSC-CM or IT MSC-CM treatment. The wound area was then measured using ImageJ softwa.

## Transwell migration assay

For the Transwell assay, $3 \times 10^4$ cells/well were suspended in medium and seeded into the upper chambers of Transwell 24-well plates (Corning, USA) with 8-μm pore filters. Then, medium with or without MSC-CM and IT MSC-CM was added to the lower chamber. After 12 h, the cells attached to the upper surface of the filter membranes were cleaned, and migrated cells on the lower surface were stained with 0.5% crystal violet for ten minutes. The level of migration was observed under an optical microscope (Motic Corporation, China).

## Real-time quantitative PCR (qPCR)

qPCR were used to measure the expression of different genes including collagen I, collagen III and bFGF by HDFs after culture with DMEM, MSC-CM and IT MSC-CM for 24h, RNA was extracted with an RNA simple Total RNA Kit (Tiangen Biotech, China) according to the instructions. The concentration of extracted RNA was measured with a microplate reader (Synergy H1, BioTek, USA). Then, RNA was reverse transcribed to cDNA using a Prime ScriptTM RT reagent kit (Takara, Japan). Real-time qPCR was performed using SYBR Green PCR Master Mix (Applied Biosystems, USA) with an ABI 7500 RT-PCR System. The expression of genes of interest (Table 1) was measured using the $2^{-\Delta\Delta CT}$ method.

## Western blot analysis

Western blot analysis was implemented as previously mentioned [6]. The primary antibodies used in this experiment are as follows: Collagen Type I (14695-1-AP, Proteintech); Collagen Type III (22734-1-AP, Proteintech). The secondary antibody was goat anti-rabbit IgG (HRP) (Signalway Antibody). The blots were analyzed using ImageJ software.

## Animal experiments

Six- to eight-week-old C57BL/6 female mice were purchased from Shanghai Slac Laboratory (Shanghai, China) and housed in the Medical Laboratory Animal Center of Jiangnan University. The Animal Ethics Committee of Jiangnan University approved all experimental procedures. Animal Ethics Approval No: JN.No20210430c0641130[113].

Preparation of acid-treated silk nanofiber (SF) solution was performed as previously described [15]. Laminin (LN) was mixed with concentrated MSC-CM or IT MSC-CM at a concentration of 3.5 μg/mL and incubated at room temperature for 30 min. Then, the above mixture was added to acid-treated SF solution to obtain SF+LN+MSC-CM or IT MSC-CM mixed solutions. The composite scaffold was obtained by freezing the mixed solution overnight at -20˚C and lyophilizing it for 72 h.

Table 1. Gene sequences used in this study.

|  | FORWARD | REVERSE |
|---|---|---|
| GAPDH | TGACATCAAGAAGGTGGTGAAGCAG | GTGTCGCTGTTGAAGTCAGAGGAG |
| bFGF | GAAGAGCGACCCTCACATCAAGC | CCAGGTAACGGTTAGCACACACTC |
| Collagen III | CTCAGGGTGTCAAGGGTGAAAGTG | TGTACCAGCCAGACCAGGAAGAC |
| Collagen I | TGATCGTGGTGAGACTGGTCCTG | CTTTATGCCTCTGTCACCCTGTTC |

Twelve C57BL/6 female mice weighing 20–25 g were implanted as follows. After the mice were anesthetized, scaffolds with or without concentrated MSC-CM or IT MSC-CM were implanted into the dorsal subcutaneous space of the mice. On the third and seventh days after implantation, the mice were euthanized via cervical dislocation, and samples were harvested for tissue hematoxylin-eosin (H&E) staining, immunohistochemistry (IHC) and immunofluorescence. The samples were fixed in 4% paraformaldehyde and subjected to IHC as described previously [16].

The tissue sections were harvested on the third and seventh days after implantation, deparaffinized and rehydrated prior to boiling in a 100˚C citrate buffer water bath for 25 minutes. The tissue sections were then washed with PBST, blocked with immunohistochemical blocking solution (Beyotime, China) for 1 hour, and incubated with primary antibodies against vimentin (Abcam, USA) and α-SMA (Proteintech, China) overnight at 4˚C. On the next day, sections were washed with PBST and incubated with the following secondary antibodies for 90 min at room temperature: Alexa 488-conjugated goat anti-rabbit IgG (Abcam, UK) and Alexa 594-conjugated goat anti-mouse IgG (Abcam, UK). The nuclei were stained with DAPI (YESEN, China). Images were acquired using a laser-scanning confocal microscope (Carl Zeiss LSM880, Germany).

## Statistical analysis

All results are expressed as the mean ± SD. The experiments were independently repeated three times. Comparisons were performed by one-way analysis of variance followed by Tukey's multiple comparison post hoc test. Statistical analysis was performed using SPSS 22.0 software. P values of less than 0.05 were considered statistically significant.

## Results

### Identification of UC-MSCs and HDFs

Surface antigen markers of P3 generation UC-MSCs were detected by FCM. The results showed that the percentages of CD31+, CD34+, CD45+ and HLA-DR+ cells were 0.00%, 0.02%, 0.1% and 0.00%, respectively. Meanwhile, the percentage of cells positive for CD29+, CD73+, CD90+ and CD105+ cells were 99.95%, 99.83%, 99.79% and 95.41%, respectively. The above results meet the standard for the identification of MSC signature antigen expression defined by the International Association for Cell Therapy. Immunofluorescence was used to identify HDFs, and the results showed that the purity of HDFs was above 95% (Fig 1A and 1B). These results indicate that the UC-MSCs and HDFs used in this study have the biological characteristics of mesenchymal stem cells and fibroblasts.

### IT MSC-CM promotes the migration of HDFs in vitro

The migration of HDFs is critical for the early phase of wound healing; therefore, we conducted scratch wound assays and Transwell migration assays to detect the migration of HDFs after treatment with DMEM, MSC-CM, and IT MSC-CM. The results showed that the migration of HDFs was higher in IT MSC-CM-treated groups than in DMEM- or MSC-CM-treated groups (Fig 2A–2D), indicating enhanced migration of HDFs by IT MSC-CM in vitro.

### IT MSC-CM promotes proliferation and activation of HDFs in vitro

HDFs proliferation and activation are critically required for rapid wound healing. Reports have shown that fibroblasts derived from patients with chronic wounds caused by diabetes and systemic immunosuppression show earlier onset of cellular senescence than fibroblasts derived

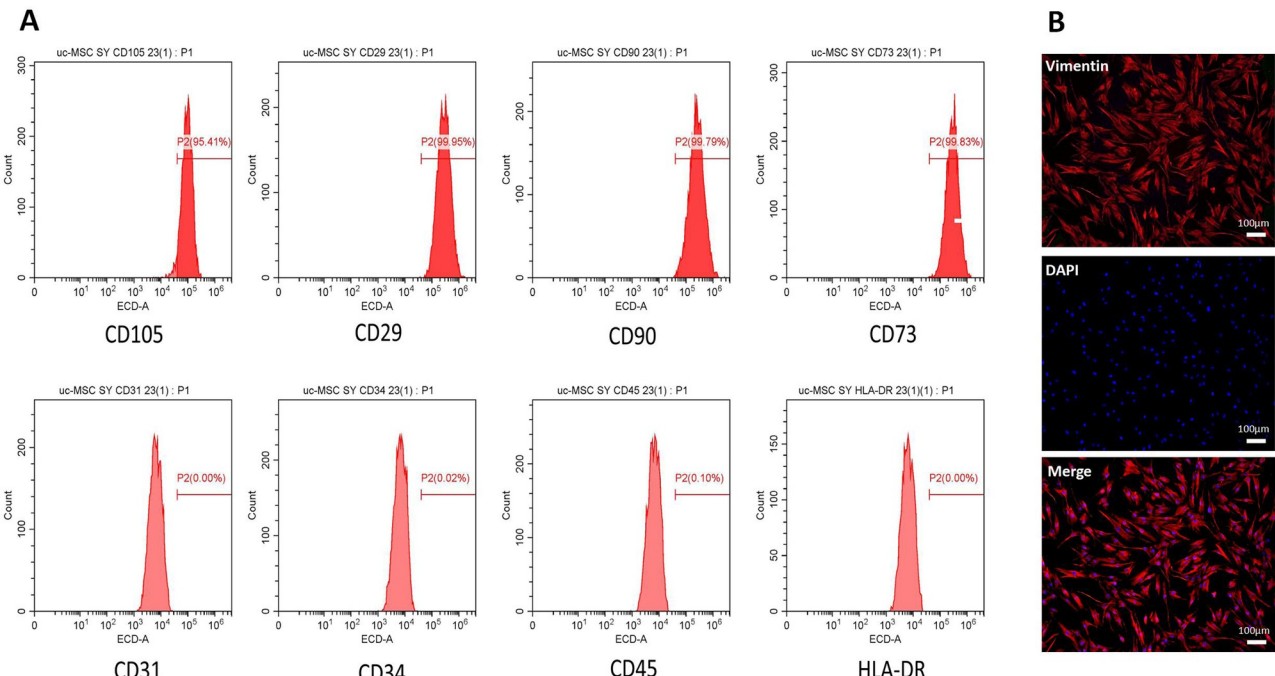

**Fig 1. Identification of UC-MSCs and HDFs.** (**A**) Results of flow cytometry identification of UCMSC surface markers: CD105+, CD29+, CD90+ and CD73+ > 95.00%; CD31+, CD34+, CD45+ and HLA-DR+ < 2.00%. (**B**) IF results of the HDF-specific protein vimentin: HDFs were positive for vimentin (red), DAPI (blue) for nuclei, scale bars: 100 μm.

from patients with healthy skin [17,18]. Cellular senescence of HDFs manifests as decreased proliferation and secretion.

To evaluate the role of IT MSC-CM in enhancing the proliferation and activation of HDFs, we detected the proliferation ability of HDFs by CCK-8 assay after treatment with DMEM, MSC-CM and IT MSC-CM. The CCK-8 results showed that cell activity was higher in IT MSC-CM-treated groups than in DMEM- or MSC-CM-treated groups (Fig 6B). In addition, we detected the activation of HDFs via IF after treatment with DMEM, MSC-CM and IT MSC-CM. Our results showed that α-SMA (green) was mroe significantly enhanced after treated with IT MSC-CM compared whit MSC-CM (Fig 3). Moreover, our qPCR results showed that the ability of HDFs to secrete bFGF was enhanced by IT MSC-CM-treated groups (Fig 4B). Furthermore, qPCR results were reconfirmed that the expression of bFGF was significantly increased in HDFs treated with IT MSC-CM (Fig 4A). In addition, compared with MSC-CM the expression of type III collagen was significantly increased in HDFs treated with IT MSC-CM, while the trend of type I was inverse in the results of qPCR and western blot (Fig 4A, 4C and 4D). We further reconfirmed the activation of HDFs was significantly enhanced after treated with IT MSC-CM. These results indicate that IT MSC-CM enhanced the proliferation and activation of HDFs in vitro.

## IT MSC-CM promotes the migration of fibroblasts in vivo

Next, we evaluated the effect of IT MSC-CM on the migration of fibroblasts in vivo. We first bound the groups of supernatant to laminin (LN) to protect their cytokine activity and then loaded the mixture into silk protein scaffold material (SF). The scaffold material SF, SF+LN+-MSC-CM and SF+LN+IT MSC-CM were implanted under the skin of the mice for 3D and

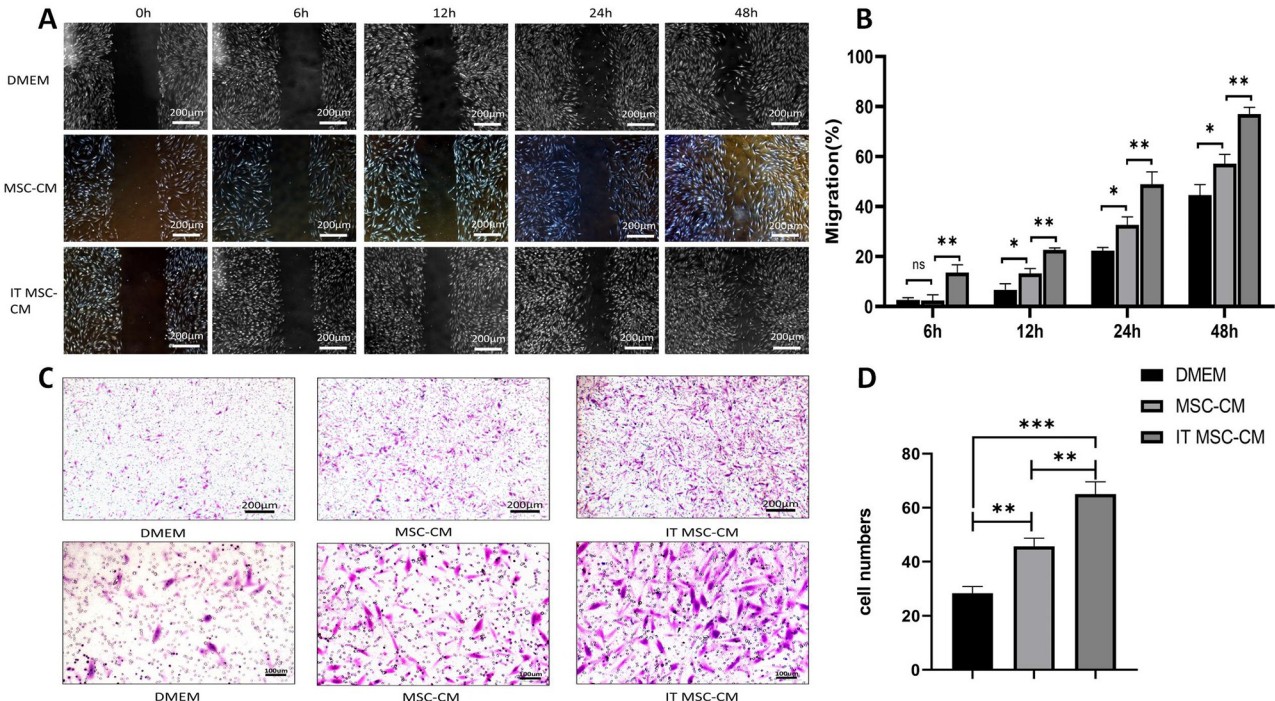

**Fig 2. IT MSC-CM promoted HDFs migration in vitro.** (**A, B**) Scratch assay of HDFs following treatment with DMEM, MSC-CM and IT MSC-CM for 2 days. Photographs were taken at 0, 12, 24 and 48 h after scratching. We calculated the migration rate using the following formula: (1-(current denuded zone area/initial denuded zone area))×100. (**C, D**) We determined the migration ability 12 h after treatment in the above groups using a Transwell cell migration assay. Cells migrating through the polycarbonate membrane were counted by detecting the average cell number in three randomly chosen fields using a light microscope. *p<0.05, **p<0.001, ***p<0.0001.

7D, respectively. The H&E results indicated that there were significantly more cells in the SF +LN+IT MSC-CM samples than in the SF and SF+LN+MSC-CM samples on both the third day and the seventh day (Fig 5A). Moreover, the immunohistochemical results showed that the cells that grew into the material were almost all vimentin-positive cells (Fig 6A), indicating that they were almost all fibroblasts.

## IT MSC-CM promotes the activation of fibroblasts in vivo

To explore the effect of IT MSC-CM on the activation of fibroblasts in vivo. The scaffold material SF, SF+LN+MSC-CM and SF+LN+IT MSC-CM were implanted under the skin of the mice for 3D and 7D, respectively. The IF results showed that α-SMA-positive cells were significantly enhanced in the SF+LN+IT MSC-CM samples than in the SF and SF+LN+MSC-CM samples on both the third day and the seventh day (Fig 7). We reconfirmed the activation of fibroblasts was significantly enhanced after treated with IT MSC-CM in vivo.

## IT MSC-CM promotes the function of HDFs may via high expression of PDGF-BB

It has been demonstrated that PDGF-BB serves as both a chemoattractant and in establishing a haptotactic gradient for HDF transmigration from peripheral tissues to the wound surface and promotes the vitality and function of HDFs [19,20]. We detected the content of PDGF-BB in IT MSC-CM by ELISA. The results showed that the concentration of PDGF-BB in IT MSC-CM was higher than that in MSC-CM (Fig 8). Moreover, studies have reported that

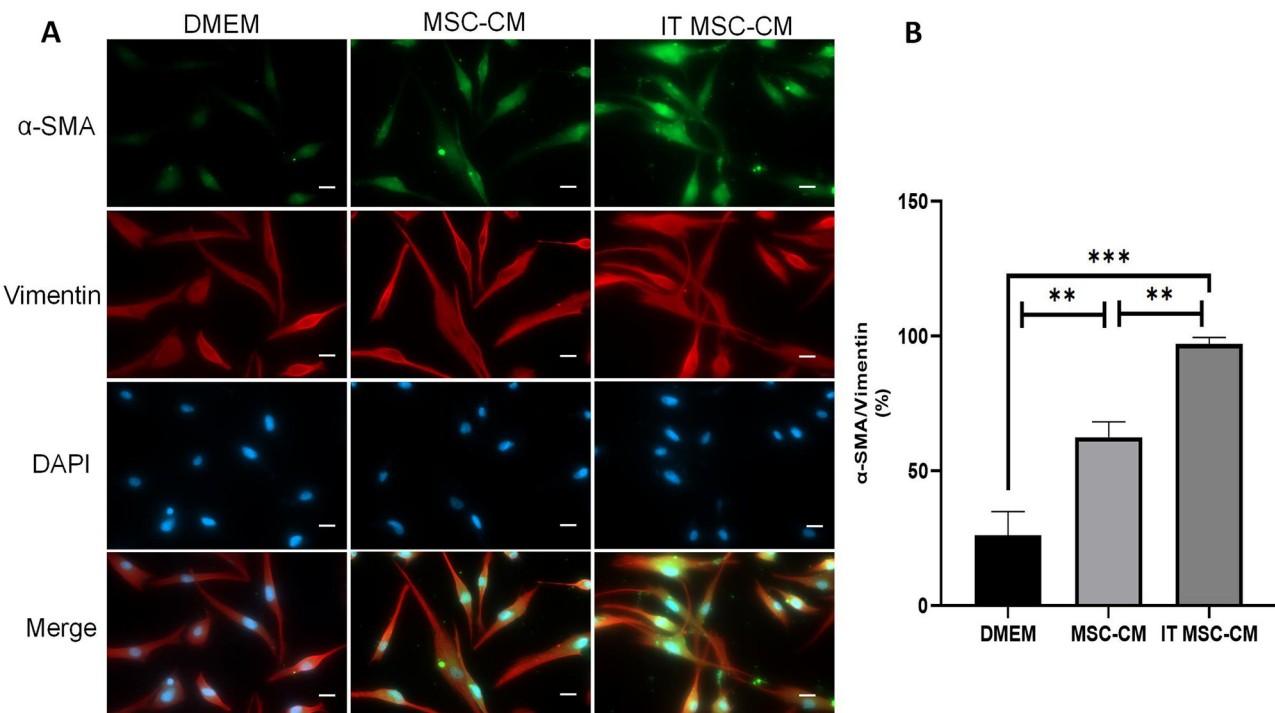

**Fig 3. IT MSC-CM promotes the activation of fibroblasts in vitro.** Immunofluorescence was used to investigate the activation of fibroblasts after 24 hours of stimulation by DMEM, MSC-CM or IT MSC-CM, respectively. (**A**) α-SMA-positive cells are stained green for activated fibroblasts, and Vimentin-positive cells are stained red for fibroblasts. DAPI (blue) for nuclei. (**B**) Corresponding histogram percentage of activated fibroblasts. scale bars: 20 μm. *p<0.05, **p<0.001, ***p<0.0001.

PDGF-BB promotes the migration of HDFs via the PI3K/Akt signaling pathway [21,22]. Therefore, we guess that the HDFs stimulated by IT MSC-CM may through high expression of PDGF-BB.

## Discussion

HDFs participate in the entire process of wound repair through the precipitation of extracellular matrix and paracrine cytokine signaling [1]. Cellular dysfunctions of HDFs, such as impaired migration, proliferation and paracrine signaling of fibroblasts, are among the key factors responsible for nonhealing wounds [5,6].

Several studies have demonstrated that the paracrine factors produced by MSCs play a critical role in wound repair [23,24]. Our study found that the paracrine function of MSCs can be amplified many times when the cells are pretreated with proinflammatory cytokines. Morever, we found that IT MSC-CM became more potent in promoting wound healing, an effect that is mediated by the upregulation of VEGFC [25].

Therefore, we speculated that pretreatment with inflammatory factors in vitro could enable MSCs to achieve a more beneficial effect on enhancing the function of fibroblasts. Indeed, our results indicate that upon stimulation by IFN-γ and TNF-α, the secretion of PDGF-BB by MSCs was greatly enhanced. Moreover, we found that IT MSC-CM became more potent in promoting the migration, proliferation and paracrine activity of HDFs, an effect that may be mediated by the upregulation of PDGF-BB.

PDGFs are a family of dimeric disulfide-bound growth factors that are important for growth, survival and function in several types of connective tissue cells [26]. Studies found that

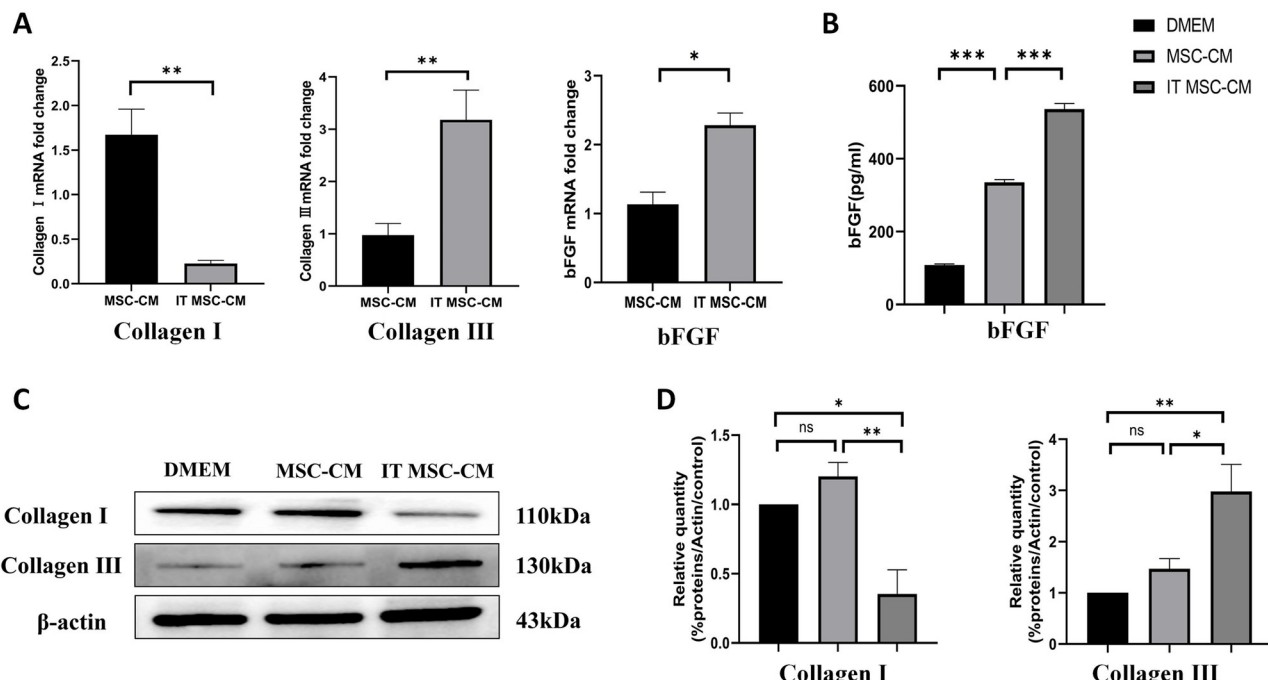

**Fig 4. IT MSC-CM improved the activation of HDFs in vitro.** (**A**) The expression of collagen I, collagen III and bFGF by HDFs were detected by qPCR after treatment with DMEM, MSC-CM and IT MSC-CM. (**B**) bFGF secretion by HDFs was evaluated by ELISA after treatment with DMEM, MSC-CM and IT MSC-CM for 24 h. Then, the cells were washed with PBS 3 times, and the supernatant was collected after adding DMEM to the dishes for 12 h. (**C**) The expression of collagen I, collagen III were detected by Western blot after treatment with DMEM, MSC-CM and IT MSC-CM. (**D**) Corresponding bar chart for Western blot. *p<0.05, **p<0.001, ***p<0.0001.

the expression levels of PDGF and PDGFR were reduced significantly in wounds of db/db mice, which was the main factor in the lack of wound healing [27]. Among these, PDGF-BB and PDGFRβ are generally accepted as having the function of accelerating wound healing [28–32]. Moreover, many studies have reported that PDGF-BB promotes the migration of HDFs via the PI3K/Akt signaling pathway [21,22]. Indeed, our in vitro study found that the migration of HDFs can effectively occur via chemotaxis in IT MSC-CM. Meanwhile, the secretion of PDGF-BB by MSCs was amplified, which indicated that the promotion of HDF migration may be mediated by IT MSC-CM via the PI3K/Akt signaling pathway.

Our previous studies have shown that acid-treated SF encourages wound healing by promoting MSCs to vascularize. However, the ability to promote cell implantation into SF remains to be improved in in vivo experiments [15]. Other studies have shown that laminin (LN) is a large glycoprotein that can effectively protect the activity of cytokines by specifically binding with a variety of growth factors and chemokines (including VEGF, PDGF, FGF, IGF, EGF, and CXCL) [33]. Therefore, under the protection of LN, SF was used as a carrier scaffold for each MSC supernatant so that the active components in the supernatant could be effectively protected. H&E staining, vimentin and α-SMA immunofluorescence of samples from our in vivo experiment showed that IT MSC-CM significantly promoted the activation and migration of fibroblasts into the SF+LN+IT MSC composite scaffold. This is consistent with the results of HDFs migration and activation experiments in vitro.

Moreover, During the trauma remodeling phase, type I and type III collagen secreted by fibroblasts are critical in promoting the precipitation of extracellular matrix on the wound surface. And a high percentage of type III collagen results in a delicate and smooth skin tissue. In

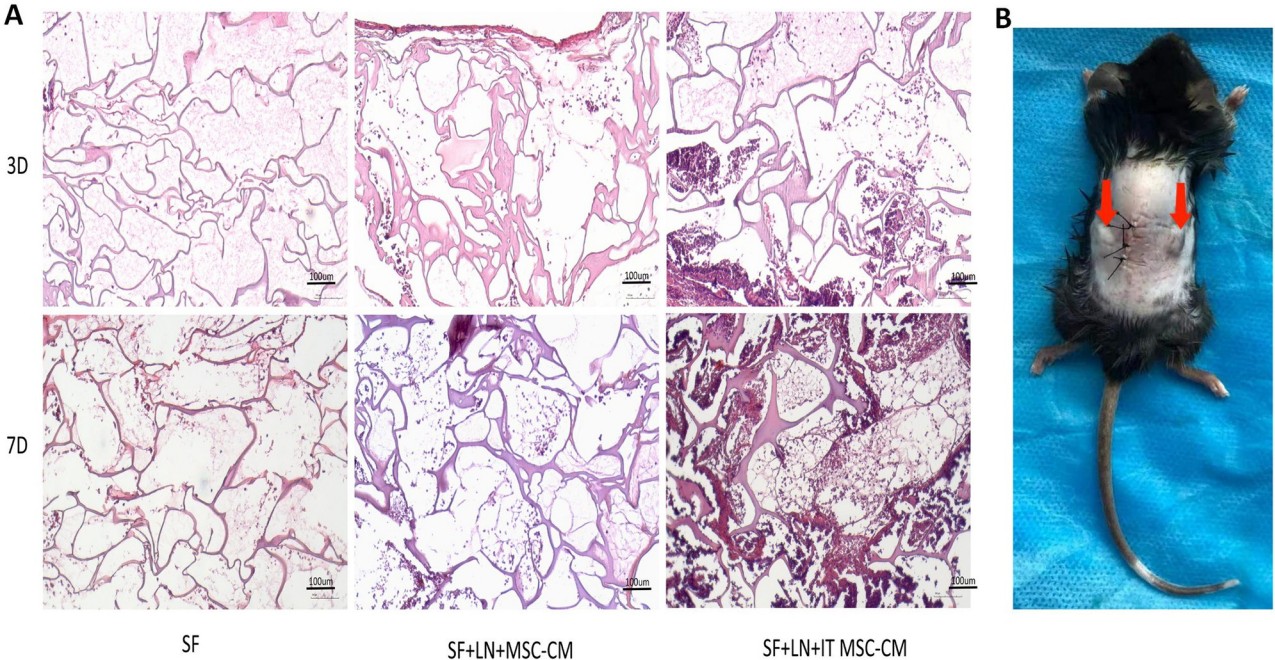

**Fig 5. IT MSC-CM promoted cells migration in vivo.** (**A**) The scaffold material specimens after implantation for 3 days and 7 days were stained with H&E to analyze the number of migrated cells. (**B**) Subcutaneous embedding photographs of mice, materials were implanted into the dorsal subcutaneous space of the mice just as photograph showed.

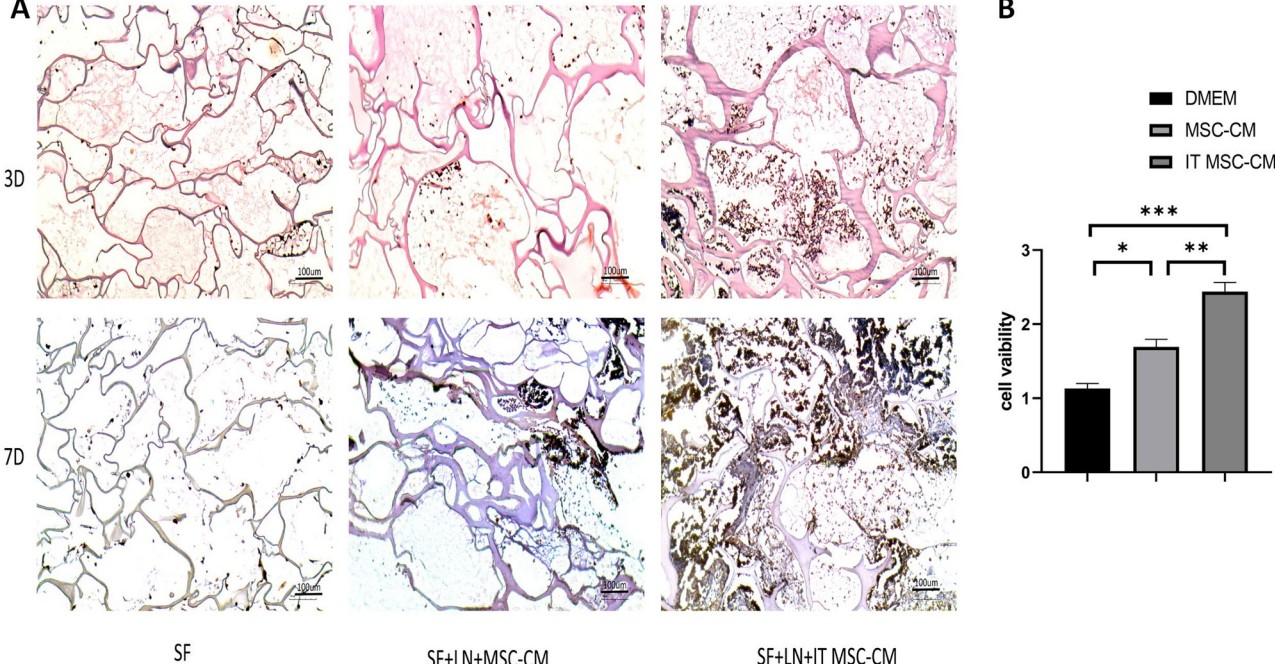

**Fig 6. IT MSC-CM promoted fibroblasts migration in vivo and the proliferation of HDFs in vitro.** (**A**) The scaffold material specimens after implantation for 3 days and 7 days were stained with vimentin antibody by immunohistochemical to detect the type of cells migrated into scaffold material. (**B**) HDF proliferation was evaluated by CCK-8 assay after treatment with DMEM, MSC-CM and IT MSC-CM for 24 h. *p<0.05, **p<0.001, ***p<0.0001.

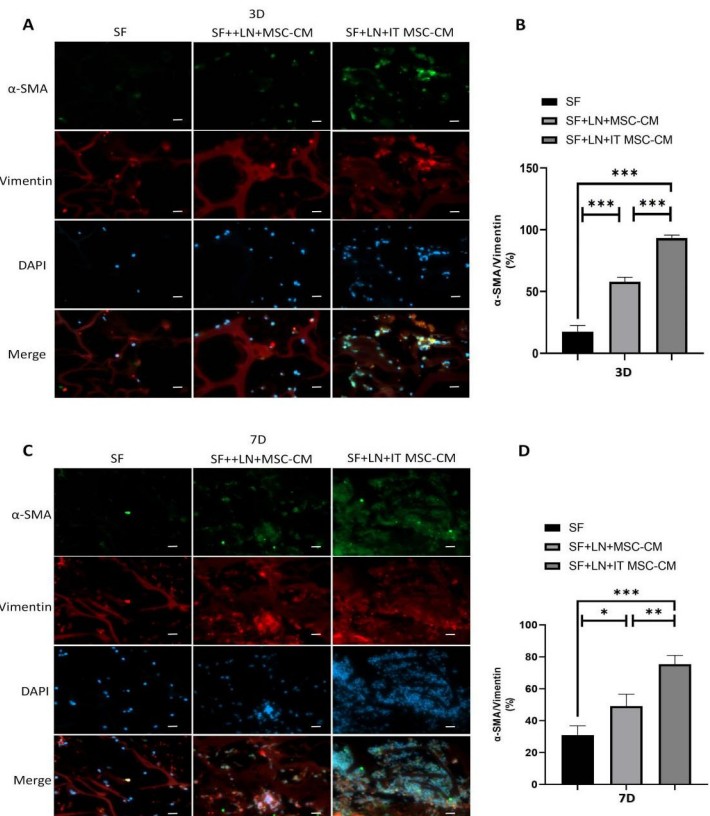

**Fig 7. IT MSC-CM promotes the activation of fibroblasts in vivo.** The scaffold material specimens after implantation for 3 days and 7 days were stained with vimentin and α-SMA antibody by immunohistochemical to detect the percentage of activated fibroblasts in scaffold material. (**A, C**) Immunofluorescence staining of α-SMA (green), vimentin (red), and DAPI (blue) in scaffold material at Day 3 and 7. (**B, D**) Corresponding histogram percentage of activated fibroblasts. scale bars: 20 μm. *p<0.05, **p<0.001, ***p<0.0001.

contrast, type I collagen appears in large quantities, resulting in a scar structure, with a hard and inelastic texture [34]. In this study, we found that IT MSC-CM significantly inhibited type I collagen expression while promoting type III collagen expression compared with MSC-CM (Fig 4), predicting that IT MSC-CM could inhibit scar formation during the trauma remodeling phase.

In summary, we found that the effects of IT MSC-CM on fibroblasts proliferation, migration and activation were superior to those of MSC-CM. These findings suggest that IT MSC-CM can be used for skin regeneration treatments and is a key factor that stimulates proliferation, migration and activation in fibroblasts, which are important for human skin rejuvenation.

## Conclusion

In this study, we found that IT MSC-CM improved the biological properties of fibroblasts, such as promoting migration, proliferation and activation in fibroblasts, and inhibited type I collagen expression and promoted type III collagen expression more than MSC-CM. Therefore, IT MSC-CM can not only promote the targeted migration of fibroblasts to the wound in a more timely manner than MSC-CM in the early stage of wound repair, but also regulate the expression of type I/III collagen during the wound remodeling phase, thus inhibiting scar

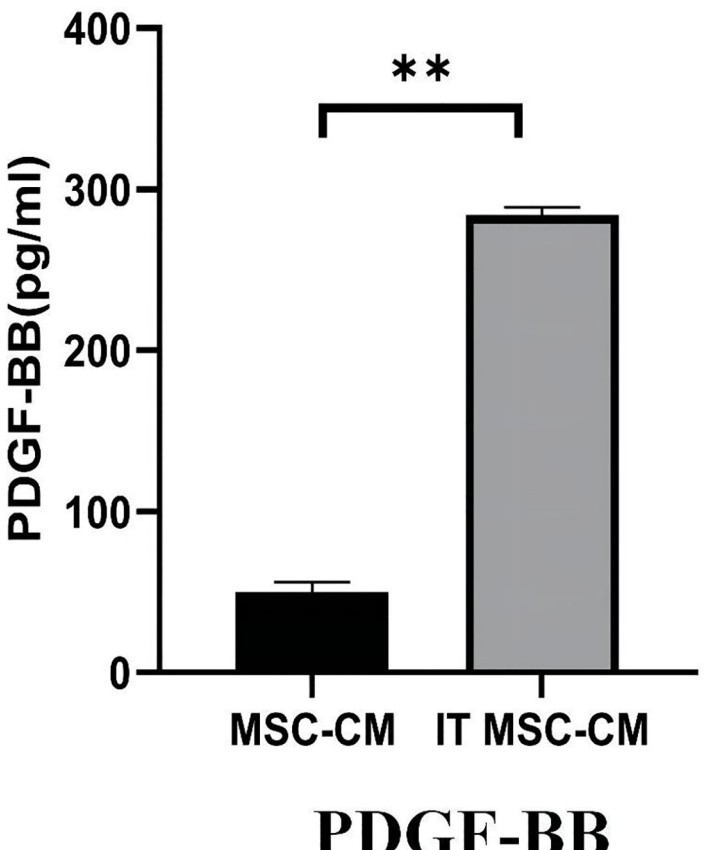

**Fig 8. High levels of PDGF-BB was detected in IT MSC-CM.** PDGF-BB secretion by MSCs was evaluated by ELISA after treatment with 20 ng/ml IFN-γ and TNF-α for 24 h. Then, the cells were washed with PBS 3 times, and the supernatant was collected after adding DMEM to the dishes for 12 h. *p<0.05, **p<0.001, ***p<0.0001.

formation. And we guess that IT MSC-CM improve the migration, proliferation and paracrine of fibroblasts maybe by high level of PDGF-BB.

## Supporting information

**S1 Fig.**
(TIF)

**S2 Fig.**
(TIF)

**S1 File.**
(ZIP)

**S2 File.**
(ZIP)

**S3 File.**
(ZIP)

**S4 File.**
(ZIP)

**S5 File.**
(ZIP)

**S6 File.**
(ZIP)

**S7 File.**
(ZIP)

## Author Contributions

**Conceptualization:** Chengchun Wang, Guozhong Lu.

**Data curation:** Kai Hu.

**Funding acquisition:** Peng Zhao, Guozhong Lu.

**Investigation:** Chenyang Liu, Chengchun Wang, Fengbo Yang.

**Methodology:** Chenyang Liu, Chengchun Wang, Fengbo Yang, Yichi Lu, Pan Du.

**Software:** Xinyao Yin.

**Supervision:** Peng Zhao, Guozhong Lu.

**Writing – original draft:** Chenyang Liu.

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
