## [Decision Letter · Decision Letter 0]

27 Jan 2022

PONE-D-21-40051The conditioned medium from mesenchymal stromal cells pretreated with proinflammatory cytokines promote fibroblasts migration and activationPLOS ONE

Dear Dr. Lu,

Thank you for submitting your manuscript to PLOS ONE. After careful consideration, we feel that it has merit but does not fully meet PLOS ONE’s publication criteria as it currently stands. Therefore, we invite you to submit a revised version of the manuscript that addresses the points raised during the review process.

This study has been carefully evaluated by two referees expert In the field. Both have judged the study of interest pending several amendments and addition of other analyses in order to make robust the results.

The Authors therefore must follow all the reviewers comments and answer to each point as well as perform the extra experiments and highlight in red color the amendments made in the text when resubmitting the second version of the paper. 

We look forward to receiving your revised manuscript.

Kind regards,

Gianpaolo Papaccio, M.D., Ph.D.

Academic Editor

PLOS ONE

Journal Requirements:

"The authors thank the National Key R&D Program of China (2016YFE0204400). We also thank the Social Development Program of Jiangsu Province (BE2018626) and Scientific Research Program of Health Committee of Wuxi City (ZM005) for support of this work."

"The authors thank the National Key R&D Program of China (2016YFE0204400). We also thank the Social Development Program of Jiangsu Province (BE2018626) and Scientific Research Program of Health Committee of Wuxi City (ZM005) for support of this work."

"The authors thank the National Key R&D Program of China (2016YFE0204400). We also thank the Social Development Program of Jiangsu Province (BE2018626) and Scientific Research Program of Health Committee of Wuxi City (ZM005) for support of this work."

Reviewers' comments:

Reviewer's Responses to Questions

**Comments to the Author**

1. Is the manuscript technically sound, and do the data support the conclusions?

Reviewer #1: Yes

Reviewer #2: Yes

2. Has the statistical analysis been performed appropriately and rigorously? 

Reviewer #1: Yes

Reviewer #2: Yes

3. Have the authors made all data underlying the findings in their manuscript fully available?

Reviewer #1: Yes

Reviewer #2: Yes

4. Is the manuscript presented in an intelligible fashion and written in standard English?

Reviewer #1: Yes

Reviewer #2: No

5. Review Comments to the Author

Reviewer #1: In this paper Authors investigated whether IT MSC-CM increases the migration, proliferation and activation of fibroblasts in vivo and in vitro.

The paper is interesting but some changes are needed.

Why was the conditioned medium removed at one time only (12h) after the treatment with IT? The authors should give an explanation for this choice or perform experiments also at different times to evaluate if the production of factors by MSCs changes and has different effects on HDFs.

Moreover, qRT-PCR should be performed also at longer times (almost 48h and 7 days). Gene expression should be confirmed by protein production.

In Paragraph 3.5 Authors refer to figure 7 and 8 but all the images are in figure 7. Please, change it.

Paragraph 3.6 should be moved before in vivo experiments, or the results should be explained in a different figure (figure 8).

Moreover, Authors assessed that “HDFs stimulated by IT MSC-CM may function through high expression of PDGF-BB”. Only ELISA assay is not sufficient to assert it, but silencing is necessary. Therefore, Auhtors should change this statement as a guess.

Reviewer #2: In this study, the Authors aimed to evaluate the effect of IT MSC-CM on fibroblasts both in vitro and in vivo. Although the paper is good, some concerns must be addressed.

First of all, molecular data must be confirmed by corresponding protein levels.

Assays of gene function loss must be performed in order to confirm the results regarding PDGF-BB expression.

In addition, the Authors should obtain conditioned medium culturing the cells at longer times. In my opinion, 12 hours are few. It would be interesting to evaluate whether there are different effects on fibroblasts due to a different amount of specific factors secreted by MSC.

The Authors must check the images showed in the figures and their description in Results section. They are not always corresponding.

6. PLOS authors have the option to publish the peer review history of their article (what does this mean?). If published, this will include your full peer review and any attached files.

Reviewer #1: No

Reviewer #2: No

---

## [Author Response · Author response to Decision Letter 0]

10 Feb 2022

Response to editor comments

Dear Editor,

Thank you very much for your letter and advice. We have revised the paper, and would like to re-submit it for your consideration. We have answered the comments raised by the reviewers point-by-point, and the amendments are highlighted in red in the revised manuscript. 

Firstly, the funding as below: This research was funded by the National Key R&D Program of China (2016YFE0204400), the Social Development Program of Jiangsu Province (BE2018626) and the Scientific Research Program of Health Committee of Wuxi City (ZM005). In additionally, The funders had no role in study design, data collection and analysis, decision to publish, or preparation of the manuscript.

Secondly, we have uploaded our study’s minimal underlying data set as a Supporting Information files.

Finally, we hope that the revision is acceptable, and I look forward to hearing from you soon.

Yours sincerely,

Chenyang Liu

Response to Reviewers

Dear reviewer:

I am very grateful to your comments for the manuscript. According with your advice, we amended the relevant part in manuscript. Some of your questions were answered below:

Part A (Reviewer 1): 

1.The reviewer’s comment: Why was the conditioned medium removed at one time only (12h) after the treatment with IT? The authors should give an explanation for this choice or perform experiments also at different times to evaluate if the production of factors by MSCs changes and has different effects on HDFs.

The author’s answer: The conditioned medium of stem cells conditioned medium was extracted by incubating stem cells after IT stimulation for 12 hours using Dulbecco’s modified Eagle’s medium (DMEM) without the nutritional support of fetal bovine serum (FBS). Considering the possible effect of starvation stress on stem cells paracrine secretion if the DMEM incubates the stem cells for too long, therefore, we used a conditioned medium that was extracted after 12h incubation.

2.The reviewer’s comment: Moreover, qRT-PCR should be performed also at longer times (almost 48h and 7 days). Gene expression should be confirmed by protein production.

The author’s answer: The conditioned medium used in this study was a DMEM extract without nutritional support from FBS. During the in vitro experiments to investigate the effect of each group of conditioned media on HDF, especially the negative control group (DMEM), considering the absence of nutritional support from FBS and necessary bioactive factors, prolonged incubation of HDF with conditioned media, starvation stress may have an effect on HDF biological activity. Therefore, conditioned medium was used to incubate HDF for 24h in this study. In addition, the protein expression corresponding to the gene expression has been added to this paper using Western blot and ELISA assays.

3.The reviewer’s comment: In Paragraph 3.5 authors refer to figure 7 and 8 but all the images are in figure 7. Please, change it.

The author’s answer: Yes, we have changed it. 

4.The reviewer’s comment: Paragraph 3.6 should be moved before in vivo experiments, or the results should be explained in a different figure (figure 8). Moreover, Authors assessed that “HDFs stimulated by IT MSC-CM may function through high expression of PDGF-BB”. Only ELISA assay is not sufficient to assert it, but silencing is necessary. Therefore, Auhtors should change this statement as a guess.

The author’s answer: Yes, We have made changes according to your suggestions.

Part B (Reviewer 2): 

1.The reviewer’s comment: Molecular data must be confirmed by corresponding protein levels.

The author’s answer: Yes, the protein expression corresponding to the gene expression has been added to this paper using Western blot and ELISA assays in Fig 4.

2.The reviewer’s comment: Assays of gene function loss must be performed in order to confirm the results regarding PDGF-BB expression.

The author’s answer: In the present study, we speculated based on ELISA results that IT MSC-CM may play a role in promoting the activation and migration of HDFs through high level of PDGF-BB, which just a guess. A more rigorous and comprehensive in vivo and in vitro validation will be carried out in the next research work of our group. We have changed the statement in this paper as a guess.

3.The reviewer’s comment: In addition, the Authors should obtain conditioned medium culturing the cells at longer times. In my opinion, 12 hours are few. It would be interesting to evaluate whether there are different effects on fibroblasts due to a different amount of specific factors secreted by MSC.

The author’s answer: The conditioned medium of stem cells conditioned medium was extracted by incubating stem cells after IT stimulation for 12 hours using Dulbecco’s modified Eagle’s medium (DMEM) without the nutritional support of fetal bovine serum (FBS). Considering the possible effect of starvation stress on stem cells paracrine secretion if the DMEM incubates the stem cells for too long, therefore, we used a conditioned medium that was extracted after 12h incubation.

4.The reviewer’s comment: The Authors must check the images showed in the figures and their description in Results section. They are not always corresponding.

The author’s answer: Yes, We have made changes according to your suggestions.

---

## [Decision Letter · Decision Letter 1]

23 Feb 2022

The conditioned medium from mesenchymal stromal cells pretreated with proinflammatory cytokines promote fibroblasts migration and activation

PONE-D-21-40051R1

Dear Dr. Lu,

We’re pleased to inform you that your manuscript has been judged scientifically suitable for publication and will be formally accepted for publication once it meets all outstanding technical requirements.

Kind regards,

Gianpaolo Papaccio, M.D., Ph.D.

Academic Editor

PLOS ONE

Additional Editor Comments (optional):

Reviewers' comments:

Reviewer's Responses to Questions

**Comments to the Author**

1. If the authors have adequately addressed your comments raised in a previous round of review and you feel that this manuscript is now acceptable for publication, you may indicate that here to bypass the “Comments to the Author” section, enter your conflict of interest statement in the “Confidential to Editor” section, and submit your "Accept" recommendation.

Reviewer #1: All comments have been addressed

Reviewer #2: All comments have been addressed

2. Is the manuscript technically sound, and do the data support the conclusions?

Reviewer #1: (No Response)

Reviewer #2: Yes

3. Has the statistical analysis been performed appropriately and rigorously? 

Reviewer #1: (No Response)

Reviewer #2: Yes

4. Have the authors made all data underlying the findings in their manuscript fully available?

Reviewer #1: (No Response)

Reviewer #2: Yes

5. Is the manuscript presented in an intelligible fashion and written in standard English?

Reviewer #1: (No Response)

Reviewer #2: Yes

6. Review Comments to the Author

Reviewer #1: (No Response)

Reviewer #2: (No Response)

7. PLOS authors have the option to publish the peer review history of their article (what does this mean?). If published, this will include your full peer review and any attached files.

Reviewer #1: No

Reviewer #2: No

---

## [Editor Report · Acceptance letter]

30 Mar 2022

PONE-D-21-40051R1 

The conditioned medium from mesenchymal stromal cells pretreated with proinflammatory cytokines promote fibroblasts migration and activation 

Dear Dr. Lu:

I'm pleased to inform you that your manuscript has been deemed suitable for publication in PLOS ONE. Congratulations! Your manuscript is now with our production department. 

Kind regards, 

on behalf of

Prof. Gianpaolo Papaccio 

Academic Editor

PLOS ONE